# Analysis of the Single Frequency Network Gain in Digital Audio Broadcasting Networks [note 1]

**DOI:** 10.3390/s21020569

**Published:** 2021-01-14

**Authors:** Kamil Staniec

**Affiliations:** Telecommunications and Teleinformatics Department, Wrocław University of Science and Technology, 50-370 Wrocław, Poland; kamil.staniec@pwr.edu.pl

**Keywords:** DAB+, MFN, SFN gain, coverage, protection level, portable, mobile

## Abstract

The single frequency network (SFN) is a popular solution in modern digital audio and television system networks for extending effective coverage, compared to its traditional single-transmitter counterpart. As benefits of this configuration appear to be obvious, this paper focuses on the exact analysis of so-called SFN gain—a quantitative effect of advantage in terms of the received signal strength. The investigations cover a statistical analysis of SFN gain values, obtained by means of computer simulations, with respect to the factors influencing the coverage, i.e., the protection level, the reception mode (fixed, portable, mobile), and the receiver location (outdoor, indoor). The analyses conclude with an observation that the most noteworthy contribution of the SFN gain is observed on the far edges of the networks, and the least one close to the transmitters. It is also observed that the highest values of the SFN gain can be expected in the fixed mode, while the protection level has the lowest impact.

## 1. Introduction

The digitization process observed in multiple areas of information and communication technologies (ICT) has also influenced broadcasting systems and services. Firstly, the terrestrial digital video broadcasting television standard, DVB-T, completely replaced its analog counterpart within the European Union. Audio broadcasting underwent a similar process, although less successfully than television due to the lack of a clear international vision concerning the future of analog audio broadcasting, hence its lower pace towards a complete switchover to a fully digital emission. The advantages presented by digitized media in the broadcasting domain greatly outperform their analog equivalents by, first of all, opening up a multitude of services unavailable in analog broadcasting suited for transmitting mainly the audio signal, with a functionally-limited radio data system (RDS) used for delivering simple text messages displayed on alphanumeric receiver screens. Another advantageous feature of digital broadcasting lies in its ability to “compress” multiple broadcasted stations (referred to as subchannels) into a single physical channel—a multiplex (MUX). To an end-listener, these subchannels would appear as separate radio stations to choose from. The range of benefits available in digital audio broadcasting (DAB) with respect to traditional analog broadcasting include (but are not limited to) the following:

Dynamic Label Segment (DLS), which allows the service provider to send text messages with supplementary information, such as track playing, now/next, news headlines, weather, sport results, etc. [1];Slide Show (SLS) for displaying information and content from traffic information to song titles, album artwork, and station branding, in both video and text formats;Electronic Program Guide (EPG) for broadcasting information on the program timetable, often in a menu-based style;Broadcast Website (BWS), giving DAB multiplex operators the opportunity to use hypertext markup language (HTML) as a content format to support information services by using the concept of a “broadcast website”. Due to this service, entire websites can be delivered to a receiver using only the broadcast channel of DAB [2];Traffic Message Channel/Traffic and Travel Information (TMC/TTI)—a technology for delivering information on traffic and travel to vehicle drivers;Emergency Warning System (EWS), which comprises several tools and functionalities available in DAB, allowing for an immediate mass-alert of DAB listeners in an area inflicted by a natural disaster. It may use a vast scope of mechanisms available in DAB such as the automatic radio receiver “wake up”, audio notification of an incoming message, instantaneous messaging, video delivery, etc.;Surround Sound over DAB+, an audio code designed especially for DAB+ to transmit multichannel sound at stereo bit rates.

### 1.1. On the State-of-the Art and the Paper Novelty

Although the concept of the single frequency network (SFN) was first used in 2007, publications regarding this issue are rather scarce and are focused chiefly on the engineering approach. In effect, there still exist vital areas that require scientific attention, such as the provision of limited-range services in wide-area SFNs; the power imbalance resulting from signal fadings due to the multipath propagation; and detailed investigation of the SFN statistics in terms of the distribution of the SFN gain values (an aspect treated in detail in this paper). Some major literature contributions relating to the aforementioned topics include a study [3] in which the average SFN gain was determined to equal 1.1 dB with a standard deviation of 3.3 dB, the values obtained for an exemplary DVB-H network located in Ghent (Belgium). In [4], a decision-directed detection was proposed for compensating the channel variation of global services, thus making the global channel estimate available on the receiver side for every OFDM symbol carrying global services. Based on this proposal, in [5,6] the authors recommended the use of a method called “local service insertion”, which allows for the provision of local services in wide-area SFNs (an option unavailable in the original version of DAB system). This method is enabled by adding an extra phase reference symbol to the DAB transmission frame. In [7], experimental SFN DVB-T2 network trials were investigated with respect to the SFN gain. The researchers proposed an empirical modification of the minimum C/N (carrier to noise) thresholds and exhaustively discussed the influence of two factors, namely the power imbalance and the relative delays (both frequently occurring in SFNs), on the performance in the Rayleigh and Rician fading channel. Similarly, an interesting attempt related to DVB-T2 (easily adaptable to DAB networks) was presented in [8], where the authors sought an optimal SFN configuration for minimal co-channel interference, proving that by carefully controlling the maximum bitrate, the reuse distance, equivalent radiated power, the inter-transmitter distances, and the effective antenna height, the network size could be remarkably extended. The fading phenomenon itself, typical to single-frequency structures, was analyzed in [9,10], in which a propagation model was developed that enabled both the signal level at the reception site as well as its phase to be determined. The model was successfully applied to a precise interference analysis of a real-life DAB+ “LocalDAB” network in Wrocław (Poland), which allowed some non-trivial conclusions to be reached regarding the depth of fades. One of the major findings in [9,10] was the observation (confirmed in field measurements) that the fades tended to be deep for receivers deployed equidistantly to a pair of transmitters while being far from the third one. However, with receivers deployed at similar distances to each of the three transmitters, the fades flattened out.

This paper’s major contribution to the subject at hand consists of providing an in-depth discussion of the SFN gain statistical distribution, performed for two scenarios:

the SFN gain analyzed over the entire coverage area;the SFN gain analyzed only in the area extending beyond the outline of the multi-frequency network (MFN) individual transmitters.

For the sake of completeness, the investigations presented by the author were carried out for a full set of parameters affecting coverage in digital audio broadcasting systems (particularly in DAB+), namely the protection layer (PL)—1A through 4A, the reception mode (RM)—fixed/portable/mobile, and the location (LOC)—outdoor/indoor. The analysis is concluded with a compilation of the SFN gain weighted average values that allowed us to notice how advantages obtained from the use of the SFN structure vary depending on the settings of {PL;RM;LOC}.

### 1.2. The Paper Organization

Section 1 introduces readers to the topic of digitized broadcasting (1.1) and presents the current state of research in the field (1.3) as well as the motivation and research novelty demonstrated in this paper. Section 2 provides an insight into the technical side of one of the leading digital audio broadcasting standards, i.e., DAB+, particularly focusing on the four coding rates, three reception modes, and two deployment locations. The rationale behind the single frequency network concept is explained in Section 3. Section 4 describes major assumptions taken while performing simulations based on a real-life DAB+ network deployment. Section 5 discusses benefits of the SFN structure in comparison to its multi-frequency network (MFN) counterpart in terms of the “SFN gain” statistical distribution. Section 6 contains conclusions and states plans for further research.

## 2. The Digital Audio Broadcasting (DAB+) System—Technical Characteristics

Unlike with DVB-T, in the area of digital audio broadcasting there exist multiple standards, which are now briefly introduced. Of primary importance in Europe, however, is the DAB+ (Digital Audio Broadcasting+) standard, whereas the other systems for digital audio include the following:

IBOC/HD Radio (officially known as NRSC-5). Developed in USA, where due to the III band occupancy DAB standard cannot be implemented. Operates by sharing a channel with the FM transmission;ISDB-TSB. Developed in Japan, introduced also in South America and in Botswana. Operates in the 470–770 MHz range on 6-MHz channels as a standard encompassing both digital television and radio, with an interface defined also for data broadcasting over several media (such as IEEE 802.3, WLAN, telephone line modem, mobile phone, etc.);DRM (Digital Radio Mondiale). An ETSI standard, initially operating in AM frequencies (<30 MHz), currently at <300 MHz (DRM30). The audio signal is mixed with small amounts of data allowing for data rates of 37–186 kb/s using the OFDM (Orthogonal Frequency-Division Multiplexing) which makes transmission considerably immune to multipath. Used in Germany, Russia, France, India, Spain, with little popularity elsewhere;DMB (Digital Multimedia Broadcasting). A standard developed in South Korea as a replacement for FM broadcasting. By offering combined services of digital audio as well as video transmissions (mobile TV) it is also a major competitor to the DVB-H (“H” for “handheld”) mobile TV standard. According to the WorldDMB Forum, DMB should be viewed as a technology for video-centric and DAB+ for radiocentric solutions.

The DAB+ standard, by far the most popular solution of those enumerated above, was developed by ETSI (European Telecommunications Standards Institute) [11,12,13]. As stated in the GE06 Agreement [14], it is intended to operate in the very high frequency (VHF) band III (174–230 MHz), i.e., in the frequency range experiencing a few-decibel (3 dB in a free space) greater propagation losses than its FM counterpart operating in the 87.5–108 MHz band. DAB+ exploits the differential quadrature phase-shift keying (DQPSK) method for modulating a multi-tone orthogonal frequency division multiplexing (OFDM) signal consisting of 1536 orthogonal subcarriers, with a total bandwidth *B* of ca. 1.54 MHz [15]. The resulting channel (MUX) capacity of 2432 kb/s allows it to multiplex up to 30 different audio stations (though up to 15 in real deployments). Thus, the composite MUX signal is transmitted in the form of a large number of bit-streams, modulating individual orthogonal physical carriers. Moreover, the insertion of a guard interval (GI) between successive OFDM symbols gives transmission a large degree of robustness against the channel selectivity caused by the multipath propagation. Out of the four transmission modes described in [12], the most suitable of these is mode I, well-fitted for terrestrial single-frequency networks in the VHF range, as it allows for greatest transmitter separations (up to 96 km). It is further recommended that the operational frequency in Mode I not exceed 375 MHz with GI = 246 μs.

Since the modulation in DAB+ is DQPSK, allowing it to transmit 2 bits per modulation symbol, the effective reception range is dependent only on the coding rate that—depending on which of the four protection levels (PL) has been endorsed—can assume the following values: 0.25 (1A), 0.375 (2A), 0.5 (3A), and 0.75 (4A).

Apart from the four protection levels (PL1-PL4), there are two more factors that influence the coverage by directly affecting the sensitivity (E_min_), one being the location, which can be either indoor or outdoor, the other being the reception mode, which can be one of the three below:

*fixed reception* defined where a directional receiving antenna is mounted on a roof, at the height of 10 m above the ground. Such an arrangement is considered to create near-optimal reception conditions, within a relatively small volume on the roof;*portable reception* defined as: class A (outdoor) which means reception where a portable receiver with an attached or built-in antenna is used outdoors at no less than 1.5 m above the ground level; class B (ground floor, indoor), which indicates reception where a portable receiver with an attached or built-in antenna is used indoors at a height of no less than 1.5 m above the floor level in rooms;*mobile reception* is when a receiver is in motion with its antenna situated at no less than 1.5 m above the ground level. This could be, for example, a vehicle receiver or handheld equipment.

## 3. The Concept and Benefits of the Single Frequency Network

The coverage area of a broadcasting station, or a group of broadcasting stations, is the area within which the wanted electric field strength is equal to or exceeds the minimum usable field strength E_min_ (i.e., its sensitivity) defined for specified reception conditions and for predefined percentage of covered receiving locations. A given area can be covered with an electric signal sufficiently strong for proper reception by means of using a single high-power transmitter. This approach, however, although efficient in open spaces or rural terrains, reveals serious flaws in urban areas, wherein tall buildings cast vast shadows, thus creating coverage gaps, as shown in Figure 1a.

Therefore, a solution better fitted for cities may be through the use of an MFN structure of the broadcasting network, in which a greater number of transmitters (Tx#1, Tx#2, and Tx#3 in Figure 1b) is used, each with a lower transmission power adjusted so that their summed serving area is the same as that intended for the original single high-power transmitter. An advantage of this arrangement is that it leads to reduction of the coverage gaps and thus to a more uniform distribution of the electromagnetic radiation across the entire area. Any remaining gaps, though smaller, can be further compensated with the use of so-called gap-fillers (or DAB-repeaters), in the form of low-power transmitters used for retransmitting signals from primary transmitters onto shadowed areas. In MFN it is assumed that all transmitters transmit the same MUX at different physical radio channels (with center frequencies f_1_, f_2_, and f_3_ in Figure 1b), capable of carrying up to several (practically not more than 15) audio or mixed audio/data streams (radio stations). In MFN each transmitter will therefore illuminate its own area, and its signal will not add up with the other transmitters’ signals, causing the received E_MFN_ to be equal to the strongest contribution arriving from any of the individual transmitters, as in Equation (1). A single-frequency network, in turn, as defined in [4], is a network of synchronized transmitters radiating the same MUX in the same physical radio channel (with the center frequency f_1_ in Figure 1c). Due to the simultaneous reception from multiple transmitters, operating at the same frequency channel, an effective coverage is thus expectedly higher by a factor known as the “SFN gain”, as defined in Equation (2) and shown in Figure 1c as a dark grey area surrounding individual coverages obtained in MFN. This gain is generated due to the fact that in the SFN configuration, individual field-strength contributions from transmitters (E_Txi_ in Equation (3), with the “i” index indicating an i-th transmitter) are combined to the total E_SFN_ by summing up, as in Equation (1). In effect the received “composite” signal can stem above the sensitivity (E_min_) required for reception, even in areas which none of the transmitters would reach individually when operating in MFN.
(1)$$E_{MFN}=\max\left\{E_{Tx1};E_{Tx2};E_{Tx3}\right\}$$
(2)$$SFN_{gain}=E_{SFN}-E_{MFN}$$
(3)$$E_{SFN}=\sum_{i}E_{Txi}$$

## 4. Numeric Simulations Based on a Real-Case DAB+ Deployment

Exact calculations of E_min_ [dBμV/m] (see [16] for the source material or [17] for a concise compilation of computational parameters) are not shown here, with only a general indication of its constituent steps involved in this process. E_min_ is found by determining the power density flux φ_min_ [dBW/m^2^] in the first step, as in Equation (4), which is a function of the noise figure NF [dB], the Boltzmann’s constant k [J/K], the reference temperature T_0_ = 297 K, the receiver noise bandwidth B [Hz], the antenna aperture A_a_ [dBm^2^], and the feeder loss L_f_ [dB]. It is then converted with Equation (5) into the minimum value of the electric field E_min_.
(4)$$\varphi_{min}=NF+10\log_{10}\left(kT_{0}B\right)+\frac{C}{N}-A_{a}+L_{f}$$
(5)$$E_{min}=\varphi_{min}+145.8$$
(6)$$E_{med}=E_{min}+\mu \sqrt{\sigma_{b}^{2}+\sigma_{m}^{2}}+L_{h}+L_{b}+L_{freq}$$

Thereupon it is scaled to the sought median value E_med_, with the use of Equation (6), by adding final adjustments accounting for factors such as the reception mode expressed by a respective value of the carrier to noise (C/N) factor; the localization (LOC), by considering corrections for the signal variability standard deviation, equal to σ_m_ = 5.5 dB (outdoors) and σ_b_ = 3 dB (indoors); the center frequency other than the reference VHF frequency f = 200 MHz (correction L_freq_); the receiving antenna height different than 10 m (correction L_h_); the building penetration losses (correction L_b_ = 9 dB); the number of locations different than 50% (accounted for by the distribution factor *μ* equal to 0.52, 1.64, and 2.33 for 70%, 95%, and 99% of locations, respectively). As for the latter, in the fixed mode, the recommended percentage of locations with the signal strength above E_min_ was accepted in [16] to be 95%, whereas for the other modes it was equal to 99% (which has its bearing on the values of the distribution factor *μ*, as stated above).

Results of these calculations are shown in Table 1 with respect to the factors that were taken into account during further analyses, i.e., four protection levels (PL = 1A–4A), two locations (LOC = {indoor; outdoor}), and three reception modes (RM = {fixed, portable, mobile}). Finally, it should be remembered that in the fixed mode, it was assumed (after [18]) that the receiving antenna is mounted on the roof (hence, no “indoor” scenario applies), and its antenna radiation pattern (ARP, shown in Figure 2) oriented towards the transmitter whose incoming signal is the strongest, which usually is the closest one, though not necessarily, depending on their equivalent isotropic radiated power (EIRP) levels. It was therefore expected that in the fixed mode the SFN gain magnitude would be constrained by this directivity, attenuating inputs from transmitters other than the dominant one. However, this assumption held true only for Rx locations in proximity to the transmitter sites, as presented in Figure 2a, where the signal contributions arriving from transmitters Tx#2 and Tx#3 fell into the ARP angular region where they underwent attenuation with respect to the ~52°-wide boresight direction (according to the stereo reception in [19], as shown in Figure 3). For more distant points, as demonstrated in Figure 2b, this angular spread was smaller, thus causing contributions from all three transmitters to fall into the Rx antenna boresight. It was calculated by the author that the latter situation concerns 86.6% of reception points lying only in the region marked as “SFN gain” in Figure 1c, whereas in the entire coverage region (i.e., in the combined area of MFN and the “SFN gain”), 62.2% of reception points would capture signals from all three transmitters within their boresight angle.

## 5. Discussion of Results—Distribution of SFN Gain Values

The results of analyses presented in this section refer to a real DAB+ network named “LokalDAB”, deployed in the SFN configuration in Wrocław (Poland) and consisting of three transmitters (sites marked as “WUST”, “IŁ-PIB”, and “Radio Wrocław”) located with respect to each other, as shown in Figure 4. For each of the resultant 24 planning configurations of {PL; RM; LOC}, simulations were performed assuming the actual effective radiated power (EIRP) from the transmitters, namely 33.83 dBm (“WUST”), 35.77 dBm (“IŁ-PIB”), and 32.70 dBm (“Radio Wrocław”).

The pathloss L_prop_ incurred by these transmissions was calculated with the use of the ITU P.1546 [20] method implemented in the “Piast” simulator [21], a tool officially used by the National Institute of Telecommunications in Poland for radiocommunication network planning purposes. “Piast” enables one to specify locations and operational parameters (including the antenna radiation pattern and a variety of correcting factors) of multiple transmitters, for pre-defined or user-defined systems, and observe the coverage obtained with the ITU P.1546 method, accounting for the terrain orography and allowing for the coverage matrix to be downloaded for further analysis, such as in this paper, where all statistics were performed on “Piast”-generated data with the use of the MATLAB environment. The scenarios were analyzed with respect to the values of the SFN gain calculated with the use of Equation (2) over the entire coverage area (Case I, shown in Figure 5a), and over only the area extending beyond the coverage outlined by the individual MFN transmitters (Case II, shown in Figure 5b). The latter scenario can be deemed as the coverage created solely due to the presence of the SFN effect in which the E-field contributions from all transmitters accumulate coherently at receivers, causing the resultant coverage to extend beyond that obtainable in MFN architecture.

Example results of the SFN gain obtained for a scenario defined by the settings {PL = 1; RM = “fixed”; LOC = “outdoor”} can be seen in Figure 6. The first observation confirmed an obvious advantage of using SFN over the MFN case in terms of the visibly increased coverage, by comparing the lower and the upper images. It could also be immediately noticed (this time returning to Figure 5) that the SFN gain was non-uniformly spread across the area. At distances near the transmitters it had lower values, increasing gradually for locations farther away, which is particularly visible in Figure 5a depicting the SFN gain across the entire coverage area (Case I). This effect can be explained by the fact that closer to a transmitter site, the signal received from it predominates contributions arriving from the other transmitters so much that their own inputs in the net received composite signal are negligible.

### 5.1. Case I—The SFN Gain over the Entire SFN Coverage

As concerns the distribution of SFN gain values, they were calculated separately for the “outdoor” and “indoor” case and presented in terms of the probability density functions (PDFs) and cumulative distribution functions (CDFs), as presented in Figure 7 and Figure 8, respectively.

As can be noticed, the most frequent occurrence (marked by the highest PDF point in Figure 7) of the SFN gain values in the fixed RM was found at around 5 dB (and existed only for the outdoor deployments, as explained in Section 4). In the portable and mobile reception, regardless of RM, the SFN gain was expected to occur at values topping at 1 dB or 2 dB, respectively. This easily observable difference between the fixed RM and the two other RMs can also be seen in Figure 8, on the CDF profiles, which are gradual for the fixed reception and by far more abrupt for the mobile and the portable scenarios, the latter particularly steep in the “indoor” case (Figure 8b). This behavior stems from the just mentioned fact that in the portable and mobile modes the SFN gain takes on rather small values (in the range of 0–2 dB), whereas in the fixed case they are most likely to be spread between 3 dB and 6 dB, as is also discussed based on detailed data in Section 6.

### 5.2. Case II—The SFN Gain in the Excess Coverage

Compared to Case I studies, the most frequent occurrences of SFN gain, in terms of the highest PDF values, could be observed in the fixed mode, achieving peak values in the form of a plateau stretched across the interval of 5–6 dB (Figure 9a). In the portable and the mobile reception modes there was a tendency for achieving spikes at lower intervals of SFN gain, namely at 2–4 dB and 1–3 dB for outdoor and indoor locations, respectively. The lowest values were again expected in the mobile RM, followed by those for the portable RM (with around 1 dB of difference between these, as presented in detail in Table 2).

### 5.3. A Detailed Discussion on Results—Major Observations

Observations and inferences drawn in this section were made based on the SFN gain values calculated for all {PL; RM; LOC} configurations, obtained separately for Case I and Case II scenarios, and compiled in Table 2. These values are in fact weighted averages of the SFN gains and are visualized in Figure 10 (Case I) and Figure 11 (Case II).

**Observation no. 1.** The SFN gain tended to grow at distances farther away from each transmitter, as can be seen in both cases shown in Figure 5 (marked as red areas). This observation substantiates the use of the SFN configuration as a recommendable design tool for improving the signal strength, particularly to users (radio listeners) located near or beyond the MFN coverage edges. It can be seen that corresponding averaged SFN gains (for outdoor reception) differed to the benefit of Case II, which was higher than its Case I counterpart by 1.4 dB in the fixed RM, and by 1.6 dB or 1.4 dB in the portable and mobile case, respectively (see column “O2-O1” in Table 2). As for the indoor reception, the weighted average SFN gains were higher in Case II than in Case I by 0.7 dB and 0.3 dB for the fixed, portable and mobile RM, respectively. These results indicate that “Case II” users (located beyond the MFN coverage), take greater advantage of the SFN gain (remembering, first of all, that without this gain they would be completely deprived of the service coverage) than the “Case I” users (located within the MFN perimeter), who would have proper signal reception anyway, even without the SFN gain.

**Observation no. 2.** A general tendency that can be observed in terms of the CDF response to the protection layer is that when the highest protection, PL = 0.25 (or 1A), is enabled, the SFN gain corresponding to the same probability *p* is higher than for the three other protection levels (2A–4A). This spread of SFN gain values due to PL attained within the same RM is the widest at *p* = 0.5 and is equals to ~1 dB in the fixed RM. The spread widens even further in the portable and mobile RM, up to about 2 dB. This phenomenon can be seen particularly well in the CDF plots in Figure 8 and Figure 12. For an explanation of this effect, one should refer to the E_med_ values in Table 1, which are the lowest for PL = 1A and the highest for PL = 4A. Hence, the amount by which the combined contribution, i.e., E_SFN_ in Equation (3), from all three transmitters exceeds the required E_med_ (or in other words—the SFN gain), will be smaller for lower-protected transmissions (i.e., with higher E_med_, such as 3A or 4A) and higher for better-protected transmissions (i.e., with lower E_med_, such as 1A or 2A), as schematically presented in Figure 13, where “E_1_”, “E_2_”, and “E_3_” symbolize signal inputs from each of the three transmitters.

**Observation no. 3.** Based on Figure 10 and Figure 11, one can notice that the expected SFN gain was the highest for the fixed reception mode, compared to the portable or mobile case, which were similar to each other. This difference, by which the average SFN gain in the fixed mode exceeded those in the other RMs, reached up to 2.4–2.5 dB compared to the portable RM and up to 1.8–1.9 dB compared to the mobile RM, with little deviations between Cases I and II (see also Table 2). Moreover, as predicted in Section 4, where the diminishing effect of the Rx antenna directivity with a distance from the transmitter sites was discussed, averaged SFN gains observed in Case II were indeed greater than those in Case I, in a measure that reflected the difference in percentages of reception points that contained signals from all transmitters within their boresight (with 0 dB attenuation) to the points where signals from non-dominant transmitters were attenuated by the ARP. As said before, at distant points of the coverage, all SFN transmitters were likely to be contained within a relatively narrow angle. This assumption held true in 86.7% and 79.1% (corresponding to PL = 1A and 4A, respectively) in Case II, which considered points only on the outermost rim of the SFN coverage. In Case I, in turn, this respective percentage was lower and equaled 62.2% and 51.8%, since this case included also points lying closer to the transmitter sites. Due to this proximity, the transmitter sites were spaced apart at an angular range larger than the antenna boresight, thus causing signals arriving from transmitters other than the dominant one to be attenuated (up to 12 dB). As can be read from Table 2, average SFN gains in Case II were greater than in Case I by up to 1.4 dB, 1.8 dB, and 1.4 dB for the fixed, portable, and mobile RM, respectively.

## 6. Conclusions and Further Research

The paper is a case study of a single frequency audio broadcasting network operating in the DAB+ standard. Advantages obtained by means of implementing the SFN configuration were analyzed with respect to the SFN gain, which was investigated in terms of its statistical distribution of values. These led to interesting results described in detail in Section 6 and summarized below for clarity:

the greatest benefit of the SFN gain is taken by users located outside the area covered by a traditional MFN network (Case II), i.e., in the rim where proper signal reception is possible only due to the combined accumulation of signals arrived from all network transmitters. As specified in Table 2, users located in this area (shown in Figure 5b) may expect an average SFN gain of up to 5.4 dB when receiving outdoors, and a maximum of 9.5 dB. Signals at receivers located inside the MFN area (Case I) also benefit from the SFN gain, though to a lesser extent, i.e., up to 4.0 dB for the outdoor reception;the SFN gain is the greatest for the fixed reception mode, differing from that achievable in the portable and the mobile RMs, on average by 1.8 dB (up to 2.7 dB at the maximum) and 2.5 dB (up to 4.0 dB at the maximum), respectively;the coding rate, expressed in the protection level, PL, has the effect of increasing the SFN gain, as PL changes from the least protected case, 4A, to the most protected one, 1A, in the range of ~0.75 dB in the fixed RM up to almost 2 dB in the portable and mobile RM;the SFN configuration causes the resultant coverage to considerably increase compared to its MFN counterpart, i.e., from 3% (in indoor mobile reception with PL = 4A) up to 54% (in outdoor fixed reception with PL = 1A). This increase, expressed in percentage, is shown in Figure 14. Again, the greatest advantage is expected for the best-protected transmissions (PL = 1A), for reasons presented in “Observation no. 3” (Section 5.3).

Further research will be devoted to the effect of the fading channels on the resultant SFN gain. The outcomes presented in this article refer to the additive white Gaussian noise (AWGN) only. The inclusion of the Rayleigh and Rician fading channels may bring about additional worth to the analysis of the SFN gain, with quite unclear and unpredictable results.

## Figures and Tables

**Figure 1 sensors-21-00569-f001:**
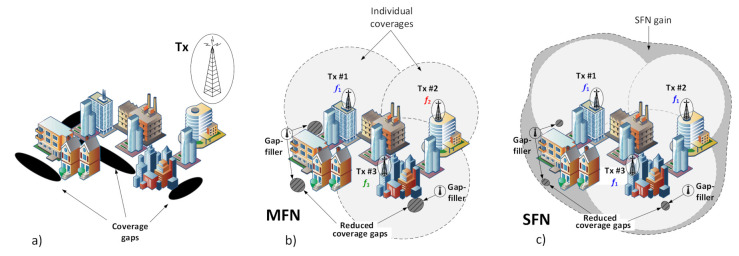
Digital audio broadcasting (DAB)+ deployment architectures: (**a**) a single transmitter; (**b**) multi-frequency network (MFN); (**c**) single frequency network (SFN).

**Figure 2 sensors-21-00569-f002:**
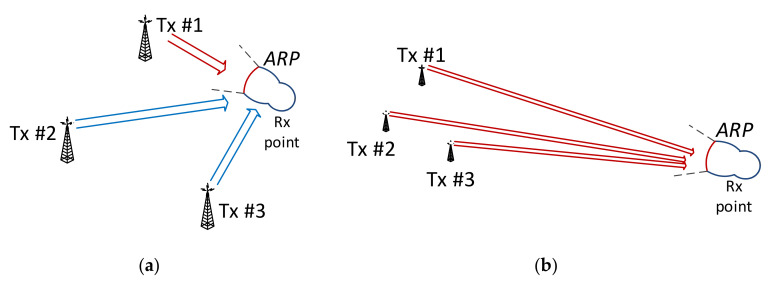
Antenna boresight with respect to SFN transmitters at close (**a**) and distant (**b**) reception points.

**Figure 3 sensors-21-00569-f003:**
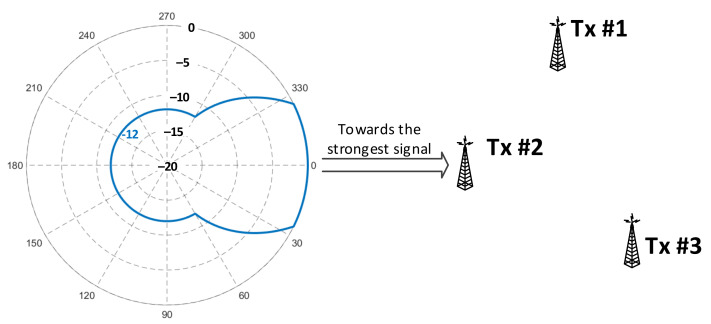
Antenna directivity (ARP) for VHF broadcasting transmission in the fixed mode [19] (angles in degrees).

**Figure 4 sensors-21-00569-f004:**
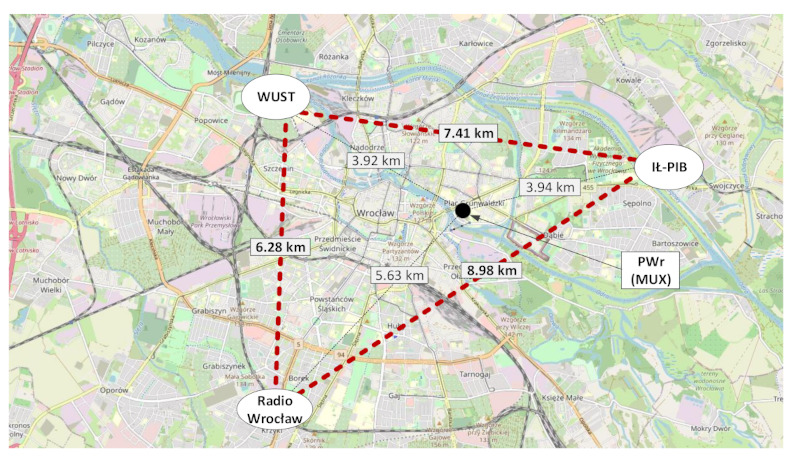
Geometry of the “LokalDAB” SFN deployed in Wrocław (map source: OpenStreetMap.org).

**Figure 5 sensors-21-00569-f005:**
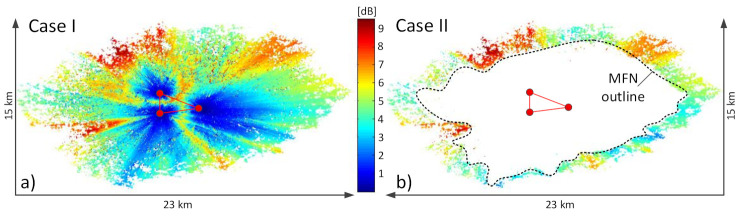
Plots of the SFN gain: (**a**) on the whole coverage area (Case I); (**b**) in the excess coverage, i.e., outside the purely-MFN coverage (Case II).

**Figure 6 sensors-21-00569-f006:**
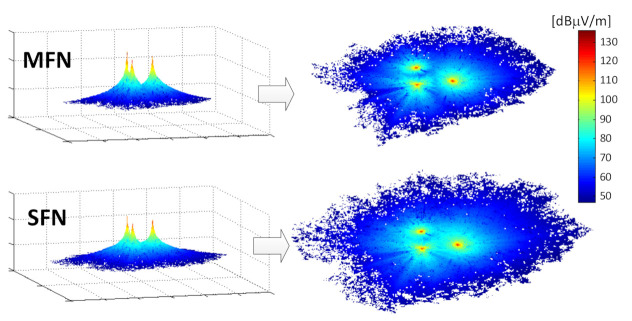
Coverage obtained in the MFN/SFN configurations and a resultant SFN gain {PL = 1A; RM = fixed; LOC = outdoor}.

**Figure 7 sensors-21-00569-f007:**
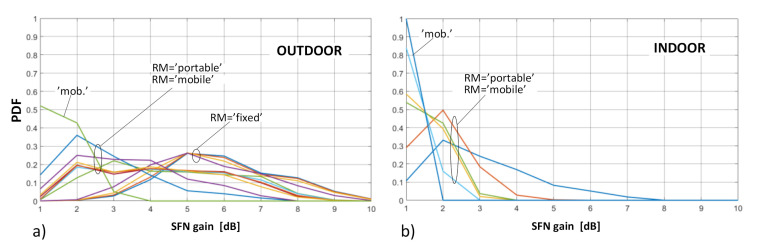
Probability density function (PDF) plots of the SFN gain distribution for different reception modes: (**a**) outdoor; (**b**) indoor.

**Figure 8 sensors-21-00569-f008:**
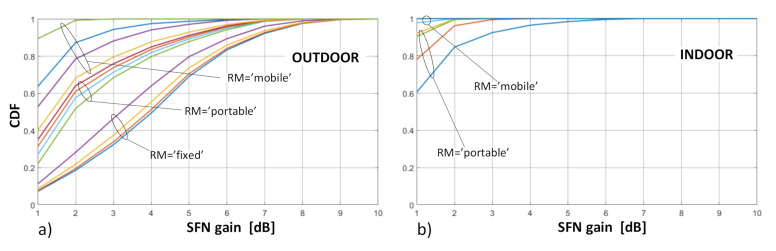
Cumulative distribution function (CDF) plots of the SFN gain distribution for different reception modes: (**a**) outdoor; (**b**) indoor.

**Figure 9 sensors-21-00569-f009:**
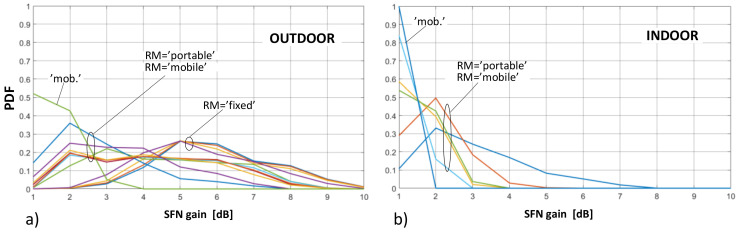
PDF plots of the SFN gain distribution for different reception modes: (**a**) outdoor; (**b**) indoor.

**Figure 10 sensors-21-00569-f010:**
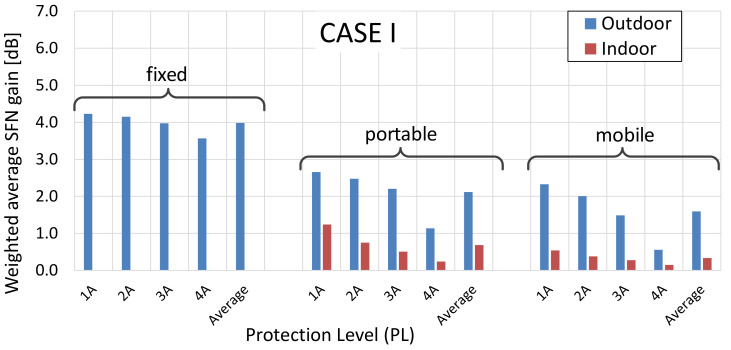
Weighted average values of the SFN gain obtained for all scenarios for Case I.

**Figure 11 sensors-21-00569-f011:**
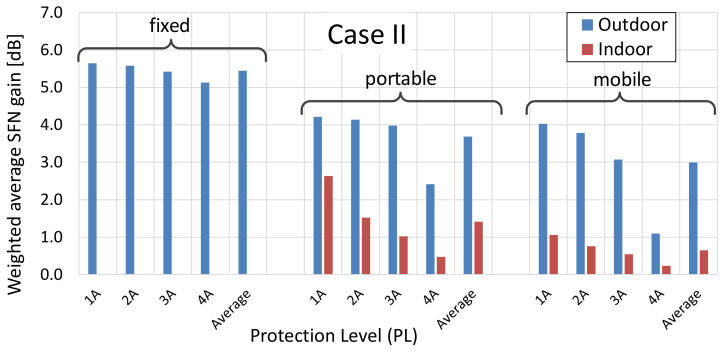
Weighted average values of the SFN gain obtained for all scenarios for Case II.

**Figure 12 sensors-21-00569-f012:**
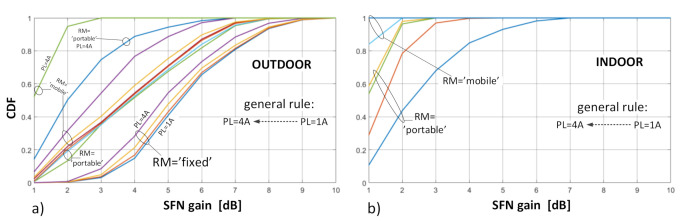
CDF plots of the SFN gain distribution for different reception modes: (**a**) outdoor; (**b**) indoor.

**Figure 13 sensors-21-00569-f013:**
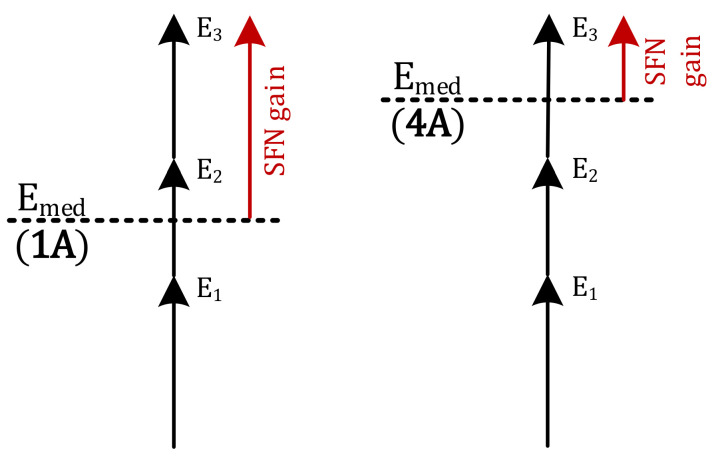
Explanation of the smaller SFN gain for different protection layers (1A and 4A).

**Figure 14 sensors-21-00569-f014:**
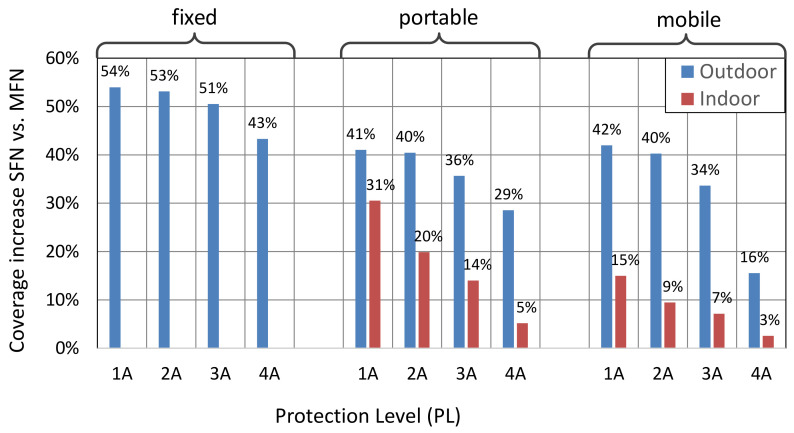
Percentage by which the MFN coverage is increased due to the use of the SFN configuration.

**Table 1 sensors-21-00569-t001:** The median electric field strength E_med_ in DAB+ for various {PL; RM; LOC} combinations (based on [16,17]).

	The Median Electric Field Strength E_med_ (dBμV/m)		
PL	Fixed Reception (Outdoor/Indoor)	Portable Reception (Outdoor/Indoor)	Mobile Reception (Outdoor/Indoor)
1A	46.7/57.0	70.9/80.8	74.7/85.2
2A	47.3/57.6	73.2/83.1	77.0/87.5
3A	48.6/58.9	75.7/85.6	79.5/90.0
4A	51.5/61.8	81.2/91.1	85.0/95.5

**Table 2 sensors-21-00569-t002:** The average (and maximum) SFN gains obtained for all analyzed {PL; RM; LOC} configurations.

RM	PL	SFN Gain (dB)				Difference in Average SFN Gain (dB)	
		Case II (Outside MFN Coverage)		Case I (Inside MFN Coverage)			
		Average (Maximum)				O2-O1	I2-I1
		Outdoor (O2)	Indoor (I2)	Outdoor (O1)	Indoor (I1)		
Fixed	1A	5.6 (9.5)	-	4.2 (9.5)	-	1.4	-
	2A	5.6 (9.5)	-	4.2 (9.5)	-	1.4	-
	3A	5.4 (9.5)	-	4.0 (9.5)	-	1.4	-
	4A	5.1 (9.5)	-	3.6 (9.5)	-	1.5	-
	Ave.	5.4	-	4.0	-	1.4	-
Portable	1A	4.2 (8.9)	2.6 (6.5)	2.7 (8.9)	1.2 (6.5)	1.6	1.4
	2A	4.1 (8.7)	1.5 (4.0)	2.5 (8.7)	0.7 (4.0)	1.7	0.8
	3A	4.0 (8.6)	1.0 (2.3)	2.2 (8.6)	0.5 (2.3)	1.8	0.5
	4A	2.4 (8.5)	0.5 (0.9)	1.1 (8.5)	0.2 (0.9)	1.3	0.2
	Ave.	3.7	1.4	2.1	0.7	1.6	0.7
Mobile	1A	4.0 (8.6)	1.1 (2.5)	2.3 (8.6)	0.5 (2.5)	1.7	0.5
	2A	3.8 (8.6)	0.8 (1.6)	2.0 (8.6)	0.4 (1.6)	1.8	0.4
	3A	3.1 (6.6)	0.5 (1.0)	1.5 (6.6)	0.3 (1.0)	1.6	0.3
	4A	1.1 (2.6)	0.2 (0.4)	0.6 (2.6)	0.1 (0.4)	0.5	0.1
	Ave.	3.0	0.6	1.6	0.3	1.4	0.3

## Data Availability

The data presented in this study are available on request from the corresponding author.

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
