# Peer review of "Analysis of the Single Frequency Network Gain in Digital Audio Broadcasting Networks â€"

_sensors, 2021, doi:10.3390/s21020569_

Round 1
Reviewer 1 Report
The paper presents a novel work as it presents a statistical analysis of SFN gain values with respect to the factors influencing the coverage, i.e. the protection level, the reception mode (fixed, portable, mobile) and the receiver location (outdoor, indoor). Below presents few areas of improvement:
- In the affiliation, you need to provide the city and country of the University.
- In the abstract, it should be mentioned that you are analysing simulation results.
- Please clearly mention about the simulator that you used to generate the results.
- the structure of the paper needs to be improved. For instance:
- section-1 should not start with a subsection-1.1. some introductory information should be there.
- subsection-1.2 should be at the end of section-1, and it does not have to have a title, it can be just the last paragraph of the introduction section.
- line-127, please fix the "" sign around radiocentric.
- Most of the references are quite outdated (older than 10 years). please add a few updated references.
Reviewer 2 Report
This is a good overview paper for the domain with a good introduction. Several places in the paper can benefit from another proofread and a little more polishing, and it would be helped by some additional references. The actual experimental outcomes are detailed well and results are believable.
Reviewer 3 Report
The paper is well written and organized. The paper analyzes the gain improvement of a DAB SFN respect to a DAB MFN. Results are based on simulations. There are three major observations from which the author extracts the paper conclusions.
The paper, however, presents just incremental results with respect to the author previous paper [15]. The novelty, if any, of the current paper respect to [15] should be clearly stated in section 1.3. Please note that if the paper has not sufficient novel contributions respect to [15], then the paper should be rejected. It should also be noted that some pictures in the current paper are exactly the same than in [15].
Regarding the study, please note that:
1.- There is not indoor fixed reception in the DAB standard. This case should be removed from the paper.
2.- According to the DAB standard, with fixed reception the coverage is for 70% of locations and μ=0.524.
3.- For fixed reception a directive antenna is used at the receiver. This antenna is supposed to be pointing toward the “best” transmitter, and would filter and reduce the signal power received from the other transmitters. The antenna pattern effect should be included in the fixed mode simulations. See ITU-R BS.599 for more information on the directive pattern.
4.- It would be interesting, for the discussion of results, to give the maximum SFN gain value that can be reached with three transmitters.
5.- For Case I results, it would be interesting to know the increase in the percentage of locations with coverage due to the use of the SFN.
6.- An explanation for the observation no. 2 should be given, as is made for observation 1.
7.- Observation 3 should be rewritten after simulations are repeated according to comment 3. SFN reception may be understood as a macro-diversity reception system, and diversity gain is higher when the combined signal strengths are similar. This is not the situation for fixed reception because a directional antenna pointing to the “best” transmitter would result in a dominant contribution from that transmitter. This third observation does not agree with the a priori expected result.
Other comments:
Acronyms should be defined just the first time they are used. There is no need for the definition of SFN in line 141 or RM in line 264.
Line 127: “radiocentric”
Lines 135 and 248. What is the meaning of c.a.? If it is “circa”, it is usually abbreviated as ca., but you can also write “around”.
Line 214: Please, use “standard deviation” instead of “standard variation”.
Figure 4: The coverage is given as power in dBm. Please, explain where is this power to be found. Is it a t the output of the receive antenna?
Round 2
Reviewer 3 Report
All previous suggestions have been correctly addressed by the authors.